# Neural encoding of perceived patch value during competitive and hazardous virtual foraging

Brian Silston[1,8], Toby Wise[2,3,4,8], Song Qi[2], Xin Sui[2], Peter Dayan [5,6] & Dean Mobbs [2,7✉]

Natural observations suggest that in safe environments, organisms avoid competition to maximize gain, while in hazardous environments the most effective survival strategy is to congregate with competition to reduce the likelihood of predatory attack. We probed the extent to which survival decisions in humans follow these patterns, and examined the factors that determined individual-level decision-making. In a virtual foraging task containing changing levels of competition in safe and hazardous patches with virtual predators, we demonstrate that human participants inversely select competition avoidant and risk diluting strategies depending on perceived patch value (PPV), a computation dependent on reward, threat, and competition. We formulate a mathematically grounded quantification of PPV in social foraging environments and show using multivariate fMRI analyses that PPV is encoded by mid-cingulate cortex (MCC) and ventromedial prefrontal cortices (vMPFC), regions that integrate action and value signals. Together, these results suggest humans utilize and integrate multidimensional information to adaptively select patches highest in PPV, and that MCC and vMPFC play a role in adapting to both competitive and predatory threats in a virtual foraging setting.

[1] Columbia University, Department of Psychology, 406 Schermerhorn Hall 1190 Amsterdam Ave., New York, NY 10027, USA. [2] Department of Humanities and Social Sciences and California Institute of Technology, 1200 E California Blvd, HSS 228–77, Pasadena, CA 91125, USA. [3] Wellcome Centre for Human Neuroimaging, University College London, London, UK. [4] Max Planck UCL Centre for Computational Psychiatry and Ageing Research, University College London, London, UK. [5] Max Planck Institute for Biological Cybernetics, Tübingen, Germany. [6] University of Tübingen, Tübingen, Germany. [7] Computation and Neural Systems Program at the California Institute of Technology, 1200 E California Blvd, HSS, 228–77 Pasadena, CA, USA. [8] These authors contributed equally: Brian Silston, Toby Wise. ✉email: Dmobbs@caltech.edu

Across phylogeny, foraging decisions (e.g. patch selection, feeding behavior, and duration) are strongly influenced by competitor density, food quantity, and expected energy cost[1–3]. In predation free, yet competitive, environments, avoiding competition dense patches is an adaptive strategy to maximize gain (e.g. see[4,5]). In contrast, foraging decisions under the potential threat of predation are governed by risk-dilution strategies (i.e. safety in numbers), for which larger groups of conspecifics reduce the chance a particular individual will fall victim to lethal attack by predators[6–9]. Risk-dilution strategies, as characterized in Hamilton's "selfish herd" theory, provide a mechanism by which prey animals aggregate and maneuver in an attempt to locate themselves between other members of the prey group, thus reducing the probability of attack by a predator, which will strike the nearest animal[10]. However, risk-dilution strategies incur efficiency costs, reducing exploitation and harvest rates[1]. Accordingly, optimal foraging theories[1,2] suggest that the conflicting trade-off between the threat of competition and the threat of predation are mitigated by the overall fitness or perceived value of the patch. That is, the subjective value should depend upon not only the level of reward or threat in the environment, but also the social context. Perceived patch value ("perceived patch value", or "PPV"), therefore, is represented as the overall potential benefit, whether in the safe domain via selection of less competition dense patches or risk dilution in the threat domain via occupation of more competition dense patches. It is thus distinct from the observed social density of a patch, as social density can result in greater or lesser value depending on the danger posed by predators. It is unclear whether humans obey these rules and whether, independent of the observable statistics of a foraging environment, is represented in the human brain.

A growing body of literature is beginning to elucidate the underlying neurobiological mechanisms of foraging decision making, and while recent work has investigated varying levels of economic risk, reward, and uncertainty[11,12], little work has included the effects of the ecological factor of threat. Human and non-human primate research has focused primarily on virtual two-patch foraging tasks in the absence of threat, consistently highlighting regions involved in action selection and value encoding (e.g. mid-cingulate cortex [MCC] and ventromedial prefrontal cortex (vmPFC)[13–15]. Further, anterior to the MCC is the dorsal cingulate cortex (dACC), which has been linked to foraging decisions and the difficulty of such decisions (see for debate[13,14]), while the vmPFC has been linked to representations of choice value during foraging tasks[14,16], which is in line with its role in monitoring action value, exploration and economic decisions[17,18]. Given the functional heterogeneity of the MCC and vmPFC, one hypothesis is that these regions reflect the perceived patch value of foraging decisions, providing a primary decision variable more immediately relevant to behavior than the simple social density of an environment. We addressed this idea by creating foraging environments that are identical except for the number of competitors in each patch. Thus, testing conditions for both safe and hazardous environments in which the optimal strategies are inversely correlated, for example competition avoidance and risk dilution, allows us to investigate the overall value of the patch independent of its directly observable statistics.

In this study, participants were scanned for approximately 4 h each over the course of 2 days while they performed a two-patch foraging task with changing level of threat and competition density (see Fig. 1A for task design). We examined multivariate, distributed neural representations involved in competition avoidance and risk dilution during a virtual foraging paradigm in which participants assessed competition density and risk of predation and made choices to enter one of two patches. First, participants learned the sequence of competitor states (specifically, the number of competitors in each patch for repeating pairs of side-by-side patches), and then were asked to select the patch in which they would like to forage. Simply selecting a patch was insufficient to receive a food reward token. Food tokens appeared at random times and locations in the patch, and the player was required to navigate to it before other competitors in the patch. Therefore, the higher the density of competitors, the less likely it was that the participant would be able to acquire the token. However, in some patches (identified by the color), a predator could appear randomly at any time or location and would capture the player or one of the other competitors, also at random.

We hypothesized that participants would adapt their foraging decisions based on the underlying perceived patch value. Perceived patch value should be higher in low social density patches during safe foraging (as a result of high reward availability). Conversely, perceived patch value should be higher in patches with greater social density when under threat of predation (greater density decreases a given individual's risk of predation), up to a level in which greatly increased competition significantly reduces resource capture. We additionally hypothesized that these decisions would be supported by neural encoding of perceived patch value independently of social density per se. Importantly, our safe and hazardous patches were matched for the effort of decision, energy costs, and competition and reward density. Further, patch switch costs were zero, therefore allowing us to investigate pure contextual changes in the perceived patch value of the decision.

The current results suggest that humans adapt decisions in response to environmental demands, varying decision preferences based on a computed perceived patch value. Here, we show that perceived patch value is associated with activity primarily in the vmPFC and MCC regions of the brain.

## Results

**Behavioral evidence for risk dilution and competition avoidance strategies.** We assessed decision making by condition by computing the percentage of trials in which the player selected the patch with fewer competitors (Fig. 1C), and in more fine-grained detail by calculating a difference score reflecting the number of competitors present in each patch at the time of decision. Participants selected the less populated patch in 89% of decisions for the safe condition and in 32% of decisions in the threat condition, inclusive of all trial durations ($\chi^2 = 4046$, $p < 0.0005$, 95% CI = [0.55, 0.58], proportion; paired $t(20) = 9.50$; $p < 0.0005$, 95% CI = [0.44, 0.69], mean of differences = 0.57). Further, we observed different average difference scores for safe and threat conditions (paired $t(20) = 9.66$, $p < 0.0005$, 95% CI = [2.53, 3.93]), indicating a context dependent shift in behavioral decision making based on perceived patch value of each patch.

Choosing patches with a greater number of competitors in the presence of potential predation increased the probability of avoiding capture as a result of risk dilution, even for the smallest patch difference (Fig. 1E; all non-paired t-test comparisons between positive and negative values $p < 0.005$). For the safe context, avoiding competition resulted in the maximum token collection (Fig. 1B). Decisions to forage in patches with fewer competitors increased capture probability compared with foraging in patches with more competitors, even for the closest value differences, e.g. between having 2 more players in the patch or two fewer players in the patch ($\chi^2 = 109.9$, $p < 0.0005$, 95% CI = [0.12, 0.19]). Players also collected significantly fewer food tokens in the threat than the safe condition, likely reflecting both increased competition and distraction due to anticipation of the predator (paired $t(20) = 13.11$, $p < 0.0005$, 95% CI = [34.64,

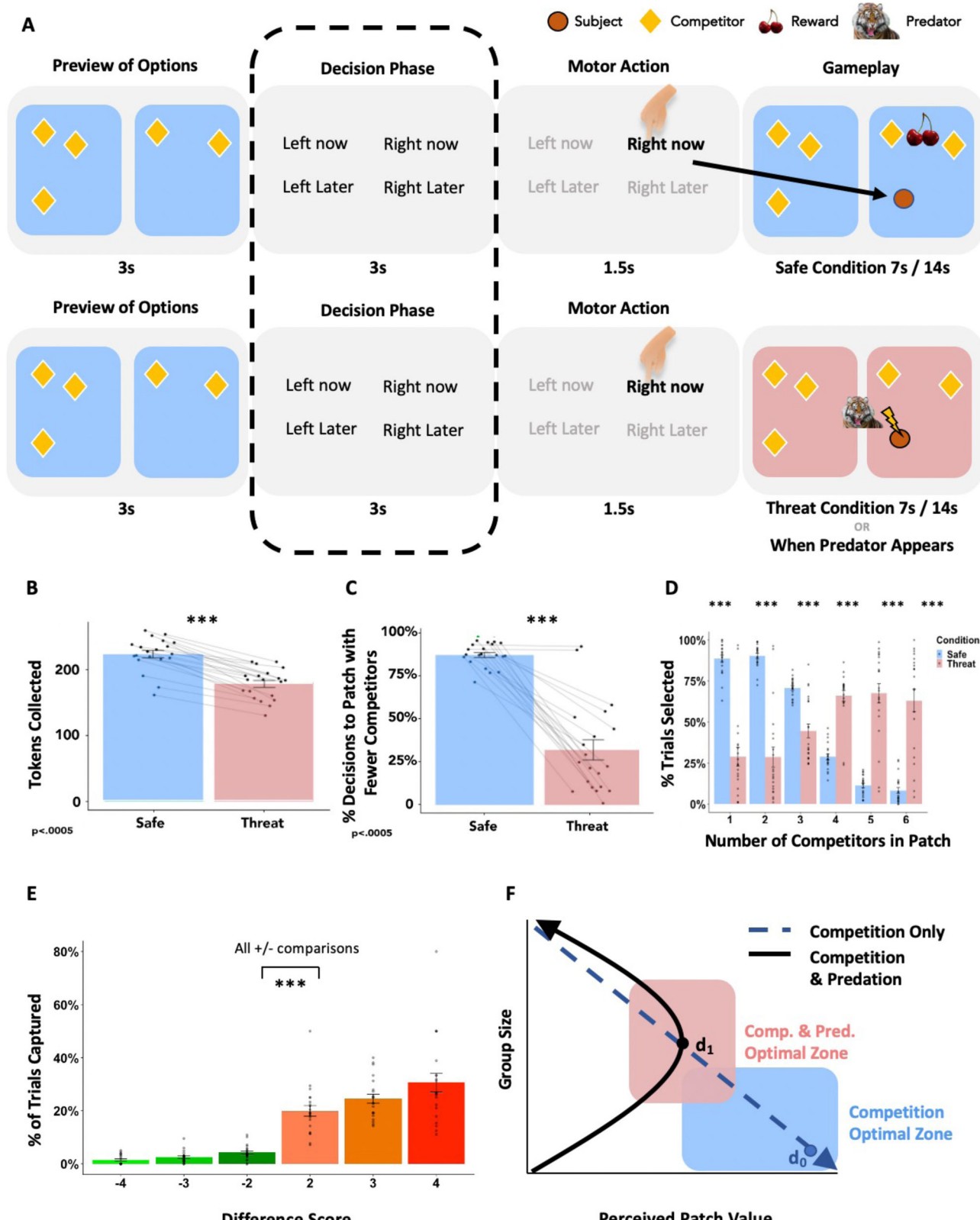

47.75]; Fig. 1B). To assess the effect of distraction of threat above and beyond basic competition we assessed mean token collection at each level of competition across both safe and threat conditions, and overall across conditions (see Supplementary Fig. 2). No discernable patterns were observed in the mean token collection as a function of number of competitors in the occupied patch.

Optimization in the safe domain via selection of less competitive patches held across all four blocks and began early in the first block, reflecting the spontaneous and swift acquisition of adaptive decision-making behavior. Likewise, most, but not all participants quickly adopted a risk-dilution strategy in the presence of predation, evidenced by consistent decision behavior across all four trial blocks. While some participants adopted an

**Fig. 1 Task design and behavioral results. A** Task Design. In the safe play (blue patches) phase the player (1) observes possible patches; (2) views the decision options; (3) makes a patch decision; and (4) is placed into the selected patch then competes to capture rewards; in the danger phase (red patches) the player follows the same procedure as safe play, except that the participant is subject to potential capture by the predator. In the safe condition the trial ends after time expires. The side-by-side patches were the same color during the experiment and are just color-coded here to clarify the differences between safe and threat conditions. **B** Players collected significantly more rewards in safe than in threat patch configurations, evidenced by a paired t-test; **C** Players spontaneously adopted a competition avoidance strategy in the safe condition, while most but not all players adopted a risk-dilution strategy in the threat condition, evidenced by a paired t-test; **D** Decision tendency as a function of threat and competition. Participants selected patches with few competitors more frequently in the safe condition, and patches with more competitors in the threat condition; all non-paired t-test comparisons $p < 0.005$. **E** Probability distribution of being captured as a function of the difference score on a given trial. A large positive difference score indicates the presence in a patch with few or one other competitor. A large negative score indicates the presence in a patch with several other competitors; all non-paired t-test comparisons between positive and negative values $p < 0.005$. **F** The risk calculus that informs individual decision making based on perceived patch value. In safe patches (blue dotted line) the optimal strategy is to select the patch with the fewest number of competitors, thus maximizing reward gain and perceived patch value. In dangerous patches (black curved line) increasing group size threatens the ability to capture rewards but dilutes risk of being the target of the predator. $d_o$ represents a decision in which perceived patch value has been maximized in the safe condition; $d_1$ represents an individual with moderate risk tolerance for both competition and threat in the danger condition, willing to select a patch with several competitors in order to reduce capture risk while still competing for rewards; Optimal Zones represent regions with high perceived patch value. *$p < 0.05$; **$p < 0.01$; ***$p < 0.005$. Pink bars represent threat condition; blue bars represent safe condition. Error bars represent SE of the mean. Sample size of $n = 21$ used to derive error bar statistics for panels. Source data are provided as a Source Data File.

identical strategy across trial type (e.g. safe/threat), most decisions in safe trials reflected a competition avoidance strategy, that is, to a patch with fewer competitors in order to maximize gain (see Fig. 1B), while the opposite was true for threat trials, on average, suggesting a risk-dilution strategy (see Fig. 1C). This pattern was also observed in finer-grained detail when examining specific numbers of competitors in each patch across conditions (Fig. 1D; all non-paired t-test comparisons $p < 0.005$).

In order to provide a mathematically grounded quantification of perceived patch value, the key decision variable emerging in our behavioral analyses, we fit a model to participants' decisions that made decisions based on the perceived patch value of each patch (see Methods). Perceived patch value depended on the average number of points collected in each condition, and included a free parameter representing the value of receiving a shock, which was negative for all but two participants (Fig. 2A). As expected, inferred perceived patch value from the model decreased with more competitors during safety and increased during threat (Fig. 2B), and the difference in perceived patch value between the two patches was strongly predictive of choice (Fig. 2C), correctly predicting 69.89% of choices. Predicted probabilities based on perceived patch value difference were also well calibrated with respect to true choice probabilities (Fig. 2D). We compared this model to variants including one that learned the value of different patches, depending on the number of competitors and threat level, and one that incorporated a tendency to stick with the previously chosen patch. Model comparison indicated that the initial model provided a better fit (WAIC = −6181.56, higher = better) versus the variant incorporating learning (WAIC = −6219.19) and the variant incorporating choice stickiness (WAIC = −9829.30).

**Medial PFC and hippocampal activity encode perceived patch value**. We next sought to determine how decision variables are encoded in the brain. We selected representational similarity analysis (RSA) to examine these variables, as this technique enables examination and comparison of representational spaces and structures deriving from different categories and data sources, e.g. neural and behavioral task data. The similitude of these spaces or structures is assessed using representational dissimilarity matrices (RDMs), which enables traditional statistical comparison across modalities.

Using RSA, we first identified regions where activity patterns aligned the task structure during the decision-making phase

of the task, during which participants were aware of available options but could not yet make a motor response. Thus, this allowed us to isolate the period where participants are evaluating the options, but not where they are making a motor action to enter their decision. Critically, RSA allowed us to identify multivariate representations of key decision variables, rather than assessing changes in single-voxel activity levels as in traditional univariate analyses, through identifying regions where neural similarity across conditions aligns with similarity in the properties of the task being performed. While univariate analyses could only determine whether perceived patch value is associated with activity in each region, RSA allows us to determine whether the multivariate representation of perceived patch value in a region differs across trials in a manner consistent with the way in which task variables differ across trials. We computed RDMs for BOLD responses to each trial using a searchlight approach, based on beta maps from a first-level general linear model including each trial as a separate regressor. This involved moving a 6 mm spherical searchlight across the entire brain and calculating the Spearman correlation between beta weights within the searchlight across every pair of trials, providing an RDM for the neuroimaging data representing the multivariate similarity between each pair of trials. The RSA analysis then sought to predict neural similarity based on the similarity of task conditions, based on the number of competitors, perceived patch value, and threat level. First, we computed task RDMs based on perceived patch value (as determined using our behavioral model) in the current patch (the patch selected on the previous trial) and the alternative. Second, we computed RDMs representing the perceived patch value of the current and alternative patch. We also included RDMs representing the (1) difference in competitors between patches; and (2) the difference in perceived patch value between patches. Finally, we included an RDM representing the (3) effect of threat. Within each searchlight sphere, we then used linear regression to predict the observed pattern of neural similarity from RDMs for task conditions (Fig. 2E), and the beta weights from these regressions provided an index of how well each task RDM predicted the neural RDM within that sphere.

This identified a distributed network of regions encoding the perceived patch value of the alternative patch during decision making, including the MCC, posterior cingulate cortex (PCC), medial prefrontal cortex (mPFC), and orbitofrontal cortex (OFC) (voxel ps < 0.05, threshold-free cluster correction, Fig. 2F). The perceived patch value of the current patch was also represented in

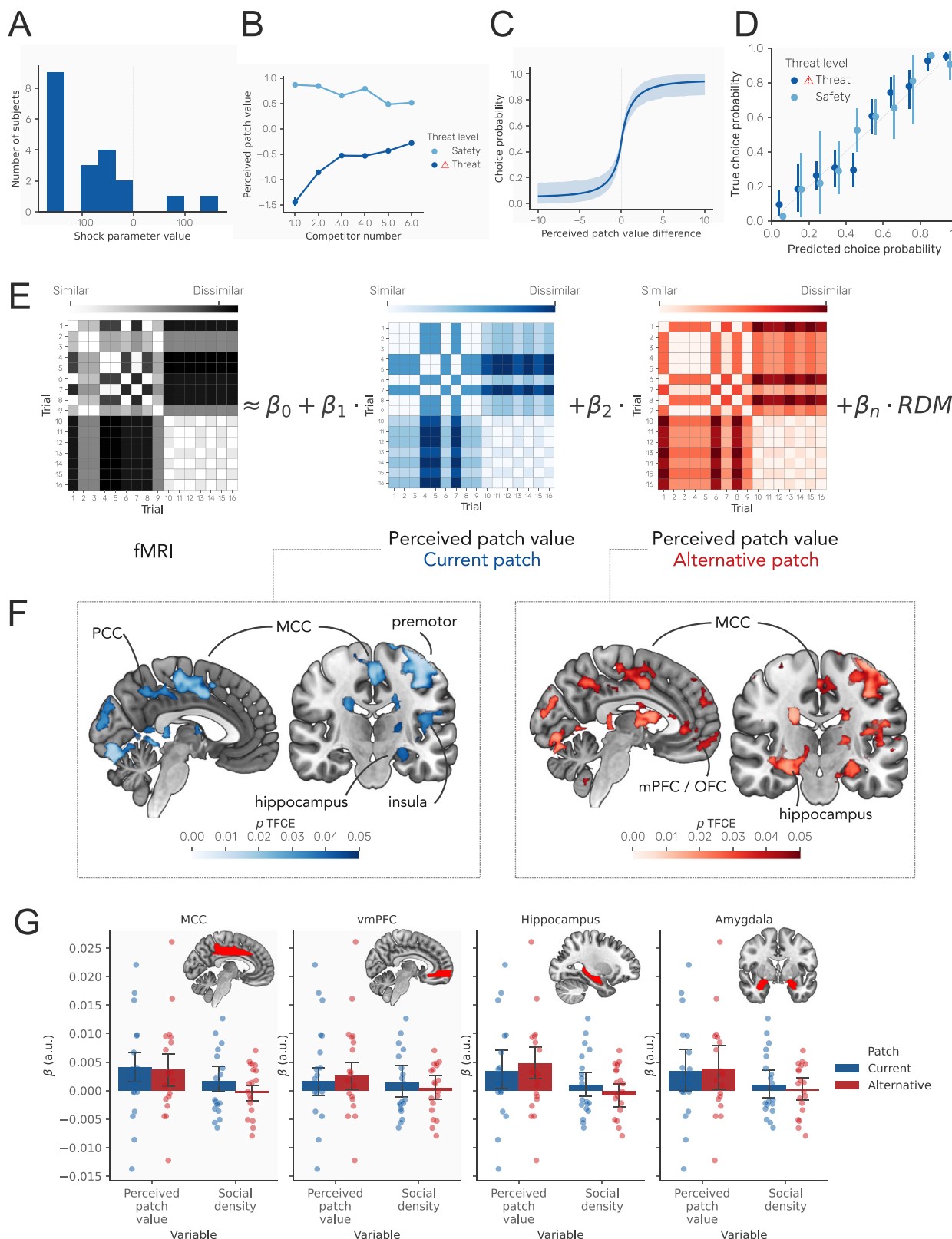

these regions (aside from the OFC), but was additionally represented in the premotor cortex, hippocampus, and anterior insular cortex (voxel ps < 0.05, threshold-free cluster correction, Fig. 2F). As shown in Fig. 2G, these regions encoded perceived patch value but did not encode social density, indicating that they are not simply encoding the number of competitors. The amygdala, a region intimately linked to threat, did not encode

perceived patch value for either patch, although there was a trend towards encoding the perceived patch value of the current patch (Fig. 2G). Importantly, our analysis approach isolated unique effects of each task RDM, controlling for effects of other task RDMs. In contrast, no areas represented the number of competitors in the current or alternative patch. Notably, we found no region encoding the difference between patches, either

**Fig. 2 Behavioral modeling and neural results. A** Values of the shock cost parameter from our behavioral model, negative for all but one participant. **B** Perceived patch value across task conditions, demonstrating that perceived patch value depends on both threat level and the number of competitors. **C** Probability of choosing a patch based on the difference between its perceived patch value and that of the alternative. The function represents the results of logistic regression models predicting choice from perceived patch value difference (see methods). **D** Calibration plot showing predicted probability of choice based on perceived patch value difference derived from a logistic regression model versus the true choice probability for safe and threat conditions. Probabilities are binned (10% bins); error bars represent 95% confidence intervals across participants. **E** Representational dissimilarity matrices (RDMs) for the neural data (hypothetical example shown left) and task conditions of interest (right). Neural RDMs modeled as a function of task RDMs using linear regression, with each RDM weighted by a weight parameter $\beta_{RDM}$. RDMs shown for a subset of trials for one participant. In the task RDMs, rows and columns represent individual trials, while the colors in the matrix represent the difference in perceived patch value between that trial and other trials. RDMs are shown for the perceived patch value of the current and alternative patch. **F** Left panel: effects of the current patch perceived patch value RDM, showing widespread effects, including the mid-cingulate cortex (MCC), posterior cingulate cortex (PCC), medial prefrontal cortex (mPFC), orbitofrontal cortex (OFC). Right panel: effects of the alternative patch perceived patch value RDM, showing effects across similar areas, with the most prominent clusters in the mid-cingulate cortex (MCC), and mPFC/OFC. Maps represent $p$ values determined using threshold-free cluster correction (TFCE), thresholded at $p < .05$, two-sided. **G** Mean extracted beta values from the MCC and vmPFC, taken from the AAL atlas[45] MCC and frontal medial orbital regions respectively, in addition to hippocampus and amygdala regions. Higher values indicate greater similarity between the task RDM and neural RDM. Values are provided for illustration only and will be weaker than actual effects due to averaging across the entire region; significance was determined using voxelwise tests as shown in (**F**). Error bars represent 95% confidence intervals across participants. All error bars and bands are calculated using $n = 19$ independent participants used for neuroimaging analyses, and the center of the error bar represents the mean. Source data are provided as a Source Data file.

in terms of number of competitors or perceived patch value. Threat level (i.e. safe or at risk of predation) was encoded in a wide range of cortical and subcortical regions (see Supplementary Fig. 3), including the MCC, vmPFC, hippocampus and amygdala.

To complement the RSA results, we also performed univariate analyses to identify regions in which overall activity levels varied according to the key decision variables involved in the task (Supplementary Fig. 4). These analyses showed that a greater number of competitors in the current patch was associated with increased activity in visual cortex, while more competitors in the alternative patch were associated with reduced activity in visual cortex. This pattern in visual cortex also emerged when looking at the effects of perceived patch value, however in addition greater value in the current patch was associated with greater activity in the thalamus, while greater value in the alternative patch was linked to greater activity in the OFC and mPFC and reduced activity in the thalamus, insula, and dorsal striatum. No significant clusters emerged when focusing on the effect of threat during the decision phase.

Finally, to facilitate comparison with prior work on value-based decision making, we performed univariate analyses focusing on the difference in value between the chosen and unchosen patch (in contrast with our primary analyses, which focused on the current and alternative patch). Results are shown in Supplementary Fig. 6, and these revealed widespread negative effects of chosen—unchosen value difference, including clusters in the dorsal anterior cingulate cortex and dorsolateral prefrontal cortex, with a single cluster in the right dorsolateral prefrontal cortex associated with the absolute value difference. Activity associated with the value of the chosen and unchosen option independently was observed primarily in occipital areas.

## Discussion
Our results demonstrate that humans adaptively select social environments based on their perceived patch value, using a competition-avoidant strategy when threat is absent and switching to a risk-diluting strategy in the presence of threat. Importantly, the key variable underpinning this decision, the perceived patch value of a foraging patch, was encoded in a network focused on the MCC and vmPFC, suggesting that these regions represent the overall perceived patch value of both the current patch and an alternative. Our results identify distributed neural systems representing key decision variables underlying adaptive foraging in response to competition and threat[19,20].

Behaviorally, we found that participants adapted their decision-making strategy based on the perceived patch value of foraging patches. Under conditions of safety participants were biased towards patches with low competition density, as found in previous work[15], representing a competition-avoidant strategy. In contrast, when under threat, participants chose patches with high social density, representing a risk-dilution strategy. These results indicate that while foraging in social environments with the threat of predation, human participants base their decisions on the perceived patch value of available patches. Different participants have different thresholds for both types of risk (Supplementary Fig. 5), suggesting a role for variability in motivational systems that may preference risk towards one domain over the other.

An important caveat of this work is that we cannot be certain how participants determine the perceived patch value of a patch. In our task, it was possible for participants to learn the value of each patch in a model-free manner, and this was reflected in our computational modeling. As a result, while the value of a patch clearly depends on the social component of the task, the participant does not necessarily need to perform any online integrative computations that adjust the value of a patch based on the level of competition, even if this may be a more effective strategy in the real world. To investigate this further, future work will need to exploit more complex experimental designs in which perceived patch value cannot be learned easily.

In the natural world, and in the absence of predation risk, immediate survival depends on maximizing food resources, and is therefore highest in environments with the fewest competitors. When under threat of predation, survival focus shifts to risk of capture, and is higher in environments with a greater number of competitors, and remains so until competition density outweighs the risk-dilution benefit. perceived patch value will depend on the interaction between multiple factors, and our task presents a simplistic case where reward is readily available. For example, when food is persistently scarce, perceived patch value is likely to be relatively high in the absence of competition even under threat.

At the neural level, we found that the perceived patch value of a social decision environment was encoded across a distributed network of regions, primarily located in the vmPFC, OFC, MCC, and PCC. These regions were largely the same for the current and alternative patch, although the value of the current patch was additionally encoded in the hippocampus, while the OFC preferentially coded for the value of the alternative patch. Notably, we found no strong evidence for encoding of perceived patch

value in the amygdala. In contrast, we found little evidence for a representation of the number of competitors per se, or the difference in perceived patch value or social density between patches.

Our use of RSA allowed us to focus on the multivariate representations of these variables in the brain, rather than relying on single-voxel activity changes. Thus, the regions we identify do not simply show similar activity levels across high perceived patch value conditions but represent survival in the same manner across conditions. Our focus on multivariate representations of decision variables during foraging, as opposed to overall activity levels, distinguishes this work from prior studies, which to date have exclusively used univariate approaches. Thus, while activity levels may be modulated by the difference between options, the pattern of activity represents the value of each option independently. This is demonstrated by comparison to our univariate analyses, which do not identify the same patterns.

While this approach does enable us to identify where in the brain these key decision variables are encoded, it provides little information about how these variables are encoded. Multivariate approaches rely on identifying distributed patterns of activity that are linked to a particular variable of interest, but it is challenging to qualitatively assess exactly what this encoding pattern looks like. For example, we are unable to say for certain that the perceived patch value is encoded relative to a particular reference value. Instead, this approach makes the implicit assumption that the variable of interest is encoded by the relative values of neural subpopulations within the region being examined. Thus, while simple coding schemes may exist within these small-scale populations, at the macro scale we cannot identify such a straightforward coding. Relatedly, there are other processes that may influence our results. For example, decision confidence and subjective value (which may be distinct from the value underpinning decisions) are likely to be confounded with perceived patch value to some degree, but were not measured in this study. Further work will be required to understand how these processes interact to influence decision making and the subjective experiences of participants during these tasks.

Historically, the MCC and vmPFC have been implicated in human foraging[14]. These regions are also primary components in our concept of perceived patch value, and active when participants consider both current and alternative patches. Importantly, our results show that these regions encode the perceived patch value of an environment across safe and threat conditions, rather than the social density of competition alone. What are these two regions computing? First, it is important to state that our MCC cluster is more posterior than the dACC area that has been linked to the difficulty of foraging decisions and the value of alternative options[8,10,13]. Also, our task conditions of interest are largely orthogonal to decision difficulty. Thus, our lack of dACC activity may reflect the matching of difficulty across conditions. Our results also indicate that the MCC is not purely signaling the value of an alternative option, or the difference between options, but simultaneously represents both the value of both the current and alternative option.[14,16,21] However, based on the knowledge of this region's connectivity and function, which suggests it plays little role in value-based, goal-directed actions per se[22], it seems incorrect to say that the MCC reflects pure value. Instead, the MCC may act as a hub, coordinating emotional responses and motor actions according to learned values[23], particularly when threat may be imminent[24]. Of note, threat level itself was also encoded in these regions, suggesting that they represent predation threat in addition to decision variables, especially in uncertain contexts[25,26].

Our results are in line with the role of the vmPFC in the valuation of options during foraging. In prior work using foraging paradigms, however, the vmPFC has been implicated as a comparator of options, while our work suggests that it independently signals the values of multiple options[27]. Others have shown that damage to the vmPFC can result in poorer learning of the value of spatial locations of rewards, supporting the role of the vmPFC in domain-general valuation[16,28]. While novel task designs are required to dissect their functional roles precisely, the extensive prior work on the roles of the vmPFC and MCC suggests that they are perfectly situated to calculate different aspects of foraging decisions based on perceived patch value and coordinate adaptive behaviors in response to threats. While the vmPFC appears important in representing the perceived patch value of options itself, potentially facilitating the decision-making process, MCC is likely involved in the coordination of motor actions (i.e. to stay or go) and has also been shown to be active in threat valence contexts[29].

While our results provide insights into the neural representation associated with aspects of the virtual foraging task, future investigations could benefit from focusing more on detailed behavioral measures. For example, because participants were given a period in which to consider their decision before having very little time to actually enter the decision (to enable analysis of the neural representation of the decision-making process), we do not have reliable access to reaction time data, which may have allowed us to investigate factors such as decision confidence. As a result of our focus on the decision-making process, the task was also not ideally suited to investigate how participants may learn the value of different patches when faced with competition and threat, and how they may make decisions that incorporate uncertainty in these learned values. This is illustrated by our modeling analyses, in which we found that a model incorporating learning did not provide as good a fit to the data, accounting for model complexity. Rewards were stable over the course of the task, and participants were able to practice the task prior to entering the scanner, allowing them to learn the value of different options prior to beginning the real task. Additionally, the length of the task (approximately 4 h) means that even if some learning does occur early in the task, the majority of the task required little learning for participants to be reasonably effective. Substantial learning or use of learned information online during the task would be evidenced by optimal decision making among current and later choice options, which was not observed at scale across participants.

The current study introduces a mathematically grounded concept of perceived patch value, which encompasses both competition avoidance and risk-dilution strategies. While not underplaying the importance of other brain regions, these strategies are represented in a network centered on the vmPFC and MCC. Importantly, these regions did not represent a computed comparative value or simple difference; instead we found evidence that perceived patch value in each patch is valued independently, and critically, independent of the number of competitors in a given patch. Individually calibrated levels of perceived patch value predicted behavioral foraging decisions in both safe and threat contexts, imbuing the concept with predictive value for a complex foraging task which, different from prior work, required assessments in competing safe and threat contexts.

## Methods

**Participants.** Participants were screened for requirements to participate, including standard health measures that determine inclusion for fMRI experiments. Following the screening, 22 (6 F; mean age: 31; range: 18–49) participants were trained on and completed a computerized, virtual foraging task featuring two conditions: safe and threat (Fig. 1A). Behavioral and neural data were lost for one session for one participant and neural data for another participant resulting in reporting of 20 participants for behavioral and neural data analyses. Participants were scanned for

approximately 4 h each over the course of 2 days. The threat condition involved the possibility of electric shock, administered to the underside of the left wrist. Electric shock intensity was individually calibrated prior to the task to an aversive, yet tolerable level. All participants provided informed consent to participate in the study, which was reviewed and approved by the Committee for the Protection of Human Subjects (IRB) at the California Institute of Technology. Subsequent to participation, all participants were debriefed regarding the purpose of the study.

**Task**. The task was a dynamic foraging game designed to investigate the effects of competition and threat on foraging decisions and behavior. Participants foraged for two, 45 min blocks per day over 2 days for a total of four blocks (Supplementary Fig. 1) and approximately 4 h of total scanning including structural scans. Each block maintained an identical trial structure, the difference being the number and cycle of competitors in the two patches. During each block a consistent cycle of competitors repeated, such that the player could quickly learn to predict the configuration (e.g. number of competitors in each patch) of the following part of the cycle. The game consisted of two patches containing 1–6 other AI players that foraged the environment for rewards, which appeared on a consistent, periodic basis.

**Cycles**. Each session was characterized by a regular progression of cycles (Supplementary Fig. 1). Each cycle consisted of three distinct competitor states that repeated sequentially throughout a session. During session one, the left patch (denoted P1) progressed in such a manner that if the participant saw one competitor during the current trial, on the following trial the participant would see five competitors, followed by two competitors. The cycle repeated for the duration of the session. The same mechanism operated for the right patch (denoted P2) such that when the participant saw one competitor in the left patch (P1), four would be present in right patch (P2); if five competitors were present in the left patch (P1), then one competitor would be present in the right patch (P2); if two competitors were present in the left patch (P1), then five competitors would be present in the right patch (P2). Different, repeating competitor state progressions were used for sessions two, three and four (Supplementary Fig. 1).

**Trial types**. Trial types included a short duration, immediate decision (SI) in which participants were briefly shown the upcoming competitor state, prompted to select either the left (P1) or right (P2) patch, were placed in the selected patch and foraged for seven seconds; short duration, later decision (SL) in which participants were briefly shown the current competitor state, prompted to select either the left (P1) or right (P2) patch for the current state in the cycle, or, if they wished, the next state in the cycle, were placed in the selected patch and foraged for seven seconds; and long duration, immediate decision (LI) in which participants were briefly shown the current competitor state, prompted to select either the left (P1) or right (P2) patch for the current state in the cycle, were placed in the selected patch, and foraged for 14 s. During the 14 s period the patch state would change once (e.g. every 7 s). Importantly, LI trials were single condition trials such that if the first 7 s was safe, the latter portion of the trial was also the safe condition, and if the first 7 s was under threat, the latter portion of the trial was also the threat condition. Since few participants utilized the later decision option (being chosen on 7.6% of trials across participants), data analyzed includes only immediate decisions categorized as SI and LI. In each of these instances the trial format was identical, and participants selected from among the patch options displayed to them prior to engaging in foraging activity, enabling us to collapse decisions across trial types. Each trial ended with a 3 s ITI.

**Decision**. At the beginning of each trial, two patches were displayed for three seconds, each with a different number of competitors (a competitor state). After the three-second period elapsed the patches disappeared, and the participant had three seconds to make a decision regarding the patch in which they would like to forage. In the SI condition, the participant chose between the two patches visually displayed, and was placed in the selected patch for the remainder of the trial. In the SL condition, the participant could choose either from among the patches visually displayed or the next iteration of patches per the repeating competitor states. If the player selected a patch from the visual display the trial would proceed identical to the SI trial type; however, if instead a patch from the next state in the cycle (not visually displayed) was selected the patches would immediately change to the following competitor configuration state and the player would proceed to forage for the remainder of the trial. In the LI condition, the patch configuration iterates halfway through the trial, forcing the player to consider both the current and next state in the cycle as they make a decision from among the visually displayed patches.

**Gameplay**. After choosing a patch, participants foraged using a diamond button box with four easy to use buttons indicating directions up, down, left and right, until trial ended, which occurred due to a time parameter or appearance of a virtual predator. AI competitors were opaque (the player could not move through AI competitors and vice versa) and programmed to chase food tokens within a

predetermined radius at the same speed as the player. During or after the threat trials the game paused for 2 s when the virtual predator appeared, after which the predator moved towards and captured a player at random, ending the trial. The predator was more likely to attack the patch in which the participant was located, but not more likely to attack the participant than other players. If the participant was captured they received an electric shock and lost a small portion of the accumulated earnings acquired over the course of the block. The trial finished with a 3 s ITI before the next trial began. Importantly, the combination of the 3 s ITI and 3 s period in which options were displayed (prior to available options on the current trial being highlighted) ensured that the decision period (the focus of the imaging analyses) was separated from the administration of shock by at least 6 s.

Safe trials consisted of making decisions based on the configuration of competition in each patch and the learned repeating cycles that enable recognition of the location within a cycle and hence, the upcoming configuration. Threat trials included the appearance of a virtual predator, adding a relevant parameter to participants' decisions. Selecting a patch with few players increases the risk of capture by a predator, but also increases the chance of acquiring rewards. Occupying a patch with several other players dilutes the risk of capture by a predator, but decreases the chance of acquiring rewards.

Each block was divided into eight (8) sub-blocks consisting of 18 trials of which half were safe trials and half were threat trials. Based on this structure, participants completed a total of 576 trials, balanced between the three trial types detailed above (SI, SL, LI). Threat trials ended when a virtual predator appeared and captured a player. The task duration over the course of 2 days was approximately 3.5 h, including a short training period. During the training period the participant observed changing patch configurations as a fixed 3-change cycle was repeated. For example, the left patch might cycle through the following number of competitors: 6, 4, 3 such that if the player observes four players in the patch, the next cycle will display three players with 100% certainty. Participants completed a post-task questionnaire, were debriefed on the experimental purpose, and paid $120 for participation.

*Behavioral data acquisition and analysis*. Participants' x–y coordinates were recorded at a sampling rate of 30 Hz. Other variables collected included reward spawn location and collection; patch selection decisions; safe/threat value of each environment, number of competitors in each environment, captures and shocks received.

Decisions tallies were generated for two focal variables including whether (1) the player selected the patch with fewer competitors; (2) the player selected a patch currently observed or a prospective patch. These decisions were analyzed based on several factors including (1) the trial type, e.g. safe or threat; (2) the trial length, e.g. short or long; (3) the block, e.g. low competitor numbers or high competitor numbers; and (4) the actual number of competitors in each patch. Data cleaning and organization was performed using Python 3.7. Statistical testing was performed using R version 3.5.0 (see below for details regarding behavioral modeling).

**Behavioral modeling**. To quantify perceived patch value at the individual participant level, we fit a model to the behavioral data. This model represented decisions as being based on the perceived patch value of the two potential patches. Perceived patch value was determined based on the average accumulated points across the task in each condition (i.e. every combination of competitor number and threat level) across the entire task, accounting for losses incurred when caught, and was calculated independently for each participant. In addition, we included the probability of being caught in each condition, multiplied by a free parameter representing the cost of receiving a shock. Thus, perceived patch value was dependent on both the number of points collected and the cost of shocks. Notably, the mean number of tokens collected across competitor number and PPV condition showed no discernable pattern, differing only three competitors at which no difference in the mean token collection was evident (Fig. S2), suggesting an effect of threat above and beyond competition vis a vis perceived patch value. The perceived patch value of condition X was therefore calculated as:

$$\text{PPV} = \text{points}_X + P(\text{shock})_A \cdot \text{shockcost} \qquad (1)$$

A softmax function with a free temperature parameter ($\tau$) was used to transform the values of the two patch options (A and B) to choice probabilities. This calculation used only the value of the two patches shown on screen, rather than later patches on LI trials, as behavioral results demonstrated little evidence for participants considering these later patches:

$$P(A) = \frac{\exp(\text{PPV}_A/\tau)}{\exp(\text{PPV}_A/\tau) + \exp(\text{PPV}_B/\tau)} \qquad (2)$$

We also evaluated a model that learned the value of patches with different numbers of competitors and threat levels over the course of the task. On each trial, the expected value of the chosen patch was estimated using a Bayesian mean tracker model. The mean ($m$) and variance ($v$) of the patch value are updated on each trial

using learning rate $G$ as follows:

$$m_t = m_{\{t-1\}} + \delta_t \cdot G_t \tag{3}$$

$$v_t = (1 - G_t) \cdot v_{\{t-1\}} \tag{4}$$

The learning rate $G$ is updated on each trial as a function of the variance and an additional free parameter theta (fixed at 1), representing the error variance.

$$G_t = \frac{v_{\{t-1\}}}{v_{\{t-1\}} + \theta_\epsilon^2} \tag{5}$$

Finally, we tested a variant of the initial decision model with an additional free parameter representing a tendency to stick with the previously chosen patch (e.g. selecting the left patch consecutively) rather than switching to the alternative. Here, the value of the chosen option from the previous trial was multiplied by this parameter, which has the effect of making it more likely to be chosen.

Models were fit in Python using PyMC3, with variational inference. Model fit was evaluated and compared using the Widely Applicable Information Criteria, which penalizes according to model complexity. For further analyses, we used the mean of the posterior distribution over parameters.

**FMRI data acquisition.** fMRI data was collected using a 3 T Prisma scanner in the Caltech Brain Imaging Center (Pasadena, CA) with a 32-channel head receive array. BOLD contrast images will be acquired using a single-shot, multiband T2*-weighted echo planar imaging sequence with the following parameters: TR/TE = 1000/30 ms, Flip Angle = 60°, 72 slices, slice angulation = 20° to transverse, multiband acceleration = 6, no in-plane acceleration, 3/4 partial Fourier acquisition, slice thickness/ gap = 2.0/0.0 mm, FOV = 192 mm × 192 mm, matrix = 96 × 96). Anatomical reference imaging employed 0.9 mm isotropic resolution 3D T1w MEMP-RAGE (TR/ TI/TE = 2550/1150/1.3, 3.1, 4.0, 6.9 ms, FOV = 230 m × 230 mm) and 3D T2w SPACE sequences (TR/TE = 3200/564 ms, FOV = 230 mm×230 mm). Participants viewed the screen via a mirror mounted on the head coil, and a pillow and foam cushions were placed inside the coil to minimize head movement. Electric stimulation was delivered using a BIOPAC STM100C.

**FMRI preprocessing.** Participants' data were preprocessed using fMRIprep[30] version stable RRID:SCR_016216[30] a Nipype[31,32] [RRID:SCR_002502] based tool. Each T1w (T1-weighted) volume was corrected for INU (intensity non-uniformity) using N4BiasFieldCorrection[33] v2.1.0 and skull-stripped using antsBrainEx-traction.sh v2.1.0 (using the OASIS template). Spatial normalization to the ICBM 152 Nonlinear Asymmetrical template version 2009c[34] [RRID:SCR_008796] was performed through nonlinear registration with the antsRegistration tool of ANTs v2.1.0[35] [RRID:SCR_004757], using brain-extracted versions of both T1w volume and template. Brain tissue segmentation of cerebrospinal fluid (CSF), white-matter (WM) and gray-matter (GM) was performed on the brain-extracted T1w using fast[36] (FSL v5.0.9, RRID:SCR_002823).

Functional data were motion corrected using mcflirt[37] (FSL v5.0.9). This was followed by co-registration to the corresponding T1w using boundary-based registration[38] with six degrees of freedom, using flirt (FSL). Motion correcting transformations, BOLD-to-T1w transformation and T1w-to-template (MNI) warp were concatenated and applied in a single step using antsApplyTransforms (ANTs v2.1.0) using Lanczos interpolation.

Physiological noise regressors were extracted applying CompCor[39]. Principal components were estimated for the two CompCor variants: temporal (tCompCor) and anatomical (aCompCor). A mask to exclude signal with cortical origin was obtained by eroding the brain mask, ensuring it only contained subcortical structures. Six tCompCor components were then calculated including only the top 5% variable voxels within that subcortical mask. For aCompCor, six components were calculated within the intersection of the subcortical mask and the union of CSF and WM masks calculated in T1w space, after their projection to the native space of each functional run. Framewise displacement[40] was calculated for each functional run using the implementation of Nipype.

Many internal operations of FMRIPREP use Nilearn[41] [RRID:SCR_001362], principally within the BOLD-processing workflow. For more details of the pipeline see https://fmriprep.readthedocs.io/en/stable/workflows.html.

**Representational similarity analysis.** First-level models were constructed using FSL 6.0[42]. Each run of the task was modeled and estimated separately, including single regressors for each task periods of no interest. Data were normalized such that each voxel had a mean value of 100 prior to analysis. As our analyses focused on the decision period of the task, we modeled this period of each trial using a separate regressor, resulting in independent beta maps for each trial[43]. Aside from the decision period, we also modeled the pre-decision period, between the trial starting and the available options being revealed, the foraging period, and the receipt of shock as events with durations corresponding to their duration in the trial, convolved with the standard hemodynamic response function. These were modeled using single regressors representing all trials across the run. We also included framewise displacement, six motion parameters, white matter and CSF

time series as regressors of no interest to account for signals related to motion and physiological processes. As the number of trials per condition was dependent on participants' choices, we excluded participants who did not experience the +/−2 or 3 competitor difference conditions. This resulted in the exclusion of a single participant.

Maps for each condition were subsequently used for representational similarity analysis (RSA) using a searchlight approach after being spatially smoothed with a 3 mm FWHM kernel. This involved calculating representation dissimilarity matrices (RDMs) for the neural data by computing the Spearman correlation distance between voxel-level representations within a 6 mm sphere for each condition across the entire task. This resulted in RDMs with as many rows and columns as trials in the task. To quantify associations between the neural RDMs and task variables, we calculated task RDMs representing the distance between conditions in terms of task features of interest.

We focused on two RDMs representing task features of particular interest: First, the number of competitors in each patch, with one RDM for the current patch, one for the alternative, and one for the difference between the two. Second, the perceived patch value of each patch, also including RDMs for current, alternative, and difference between patches. We then used linear regression to determine the influence of task RDMs on neural RDMs, with each RDM being weighted by its own β parameter. We also included a further RDM representing threat level, along with RDMs representing condition similarity in run number (where conditions from the same run have the highest similarity) and session number (where conditions from the same session have the highest similarity). This produced whole-brain maps representing the β weights for the RDMs of interest, with each voxel representing the effect in the 6 mm sphere of which it was the center.

Maps from the searchlight RSA analysis were thresholded using the randomize function in FSL, using a one-sample t-test against zero. Statistical significance at every voxel was determined using threshold-free cluster correction with 5000 permutations and 10 mm variance smoothing. While statistical significance was determined using this whole-brain approach, for the purposes of illustrating the pattern of effects across different regions, we extracted beta values from the RSA within key regions showing significant effects (as shown in Fig. 2). For this we used the AAL atlas, selecting the MCC, frontal medial orbital cortex, hippocampus and amygdala regions, extracting the mean beta value from the RSA within the region for each participant. We show the mean and 95% confidence intervals in Supplementary Fig. 3.

**Univariate analysis.** Univariate analyses were conducted using Lyman (http:// www.cns.nyu.edu/mwaskom/software/lyman/index.html). Data were first smoothed using an 8 mm FWHM kernel, and first-level models were constructed as described for the representational similarity analysis for each run. The only exception was that rather than modeling the decision period of each trial sepa-rately, we instead modeled the decision period using a single regressor for each trial, including parametric modulators for each decision variable of interest (per-ceived patch value and the number of competitors for the current and alternative patch, the difference between these values, and threat level). No participants were excluded for this analysis. Maps from first-level analyses for each run were then combined using a second-level fixed-effects analysis within participant, and participant-level maps from this stage were finally used for group-level analysis. At the group level, significance was determined using FSL's randomize tool, with 5000 permutations and 10 mm variance smoothing. Results of the univariate analysis are displayed in Supplementary Fig. 4.

Additional analyses focusing on the value difference between chosen and unchosen options were conducted using the same approach, including regressors representing the value of the chosen option, the value of the unchosen option, the value difference between chosen and unchosen options, the absolute value difference between chosen and unchosen options, and the sum of the value of the two options. These were modeled in separate GLMs due to collinearity between regressors.

**Reporting summary.** Further information on research design is available in the Nature Research Reporting Summary linked to this article.

## Data availability

The behavioral data generated in this study have been deposited on the GitHub[44] repository (https://github.com/mobbslab/foraging_paper) and Zenodo (https://doi.org/ 10.5281/zenodo.5113171). The processed neural data are available at the Open Neuro database, at https://openneuro.org/datasets/ds003484. Raw neural data will be made available upon request to the corresponding author. Source data are provided with this paper.

## Code availability

All code used for the analysis and modeling of behavioral and neural data in this study is available at GitHub (https://github.com/mobbslab/foraging_paper) and Zenodo (https:// doi.org/10.5281/zenodo.5113171).

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

## Acknowledgements

This work was supported by the National Institute of Mental Health grant 2P50MH094258 and 1 R01 AG043463-01, Chen Institute award P2026052, Templeton Foundation grant TWCF0366, Sir Henry Wellcome Fellowship 206460/Z/17/Z, the Max Planck Society and Humboldt Foundation.

## Author contributions

The study was conceptualized and designed by B.S., D.M., X.S., and P.D., and created by B.S. Material preparation, data collection, and analysis were performed by B.S., S.Q., X.S., and T.W. The first draft of the manuscript was written by B.S., D.M., and T.W. and all authors commented on previous versions of the manuscript. All authors read and approved the final manuscript.

## Competing interests

The authors declare no competing interests.
