## [Peer Review File · Nature Communications]

Neural encoding of perceived patch value during competitive and hazardous virtual foragingREVIEWER COMMENTS

Reviewer #1 (Remarks to the Author):

The authors report an analysis of a new foraging paradigm in which participants are asked to choose a foraging patch while considering the density of competing agents and, under one condition, a predator that leads to a loss of tokens, a shock, and the end of the trial. Participants performed the task during functional magnetic resonance imaging (fMRI). After confirming that participants responded to the primary variables manipulated in the task (competitor density and predator threat), the authors construct a computational model of behavior modeling a concept called “socially adjusted value” which they show to be affected by points collected, probability of being shocked, and distaste for being shocked. The authors then test for drivers of variation in RDM matrices to identify areas of the brain that make the targeted distinction. They find a correlation for socially-adjusted value with putative value regions of the brain along with midcingulate cortex, posterior cingulate, and hippocampus and insula for the current patch. Representations for threat were found in similar regions. The authors conclude that a socially adjusted representation of value that incorporates threat and density tradeoffs is represented in a distributed network of brain regions.

The task is fun and thought provoking and the significant data collected for participants makes the paper more interesting as well as a set of complex analysis more feasible.

Larger concerns:

1. During the safe condition, it is clear from Fig S# that participants are still achieving high rates of reward with a larger number of competitors. It seems plausible that this is accomplished through faster response times and/or a greater number of key presses (though the details described are insufficient to be sure). Although there is a clear bias to choose the less populated patches, the choice is not absolute, this might be the reason. In general, the safe condition with more participants sounds just as rewarding and perhaps more engaging. If a larger number of movements (or more complex plans) are made it could explain the RSA result in the MCC as well as the widely distributed separability that hints at general motivation differences.
2. Please add a description of the specification of ROIs. A quick reference to AAL appears but what was done was not clear from the methods. This also makes it difficult to compare these regions to the specific ROIs referenced in the manuscript.
3. Degrees of freedom in t-tests change for similar tests without explanation.
4. It is very challenging to review and have others replicate or extend work that is only available on request. Please: 1. Make all analysis scripts (main stats, SI, Neuroimaging, RSA) available on GitHub or a similar repository. 2. Make minimally processed fMRI data available on OpenNeuro or similar.
5. Need to see a correlation table (or list) between RDMs. Are they all independent?

Minor:

- line 27 “than” -> that
- line 51 “decoupled” usage unclear.
- Social adjusted value always appears in separate font
- line 93 parenthetical contradicts immediately preceding statement without explanation.
- line 111 “The foregoing tendency to congregate in numbers in the presence...” is difficult to parse/understand
- line 188 “based [on?] socially adjusted value”
- line 191-2 Difficult to understand. This is three things, right?
- line 205 “value” - disambiguate?
- line 315 comma precedes citations at the end of the sentence
- line 325-6 font change.

- line 337 "contexts imbuing the concept" - ? imbuing?
- line 424-5 line break
- line 485 "will employ" - tense

Reviewer #2 (Remarks to the Author):

The authors developed a naturalistic foraging task to measure how people evaluate competition during foraging with or without predation: When the environment is safe, an agent should want fewer competitors, but under threat, competitors offer safety from a predator. The authors used fMRI to look for multivariate representations of the value signals involved in making this choice. They found that participants adjusted their strategy depending on the threat context, and observed context-dependent value signals in ventromedial PFC and (posterior) mid-cingulate, suggesting that these regions are involved in integrating information about potential competition and threat to determine stay-leave decisions.

The topic of how people make decisions in dynamically changing social environments, and its connections to real-world foraging behavior, is of great interest to researchers across several disciplines. These authors have taken a novel approach to addressing this, using a cleverly designed task and substantial amounts of within-subject data. However, while I think this study has the potential to confirm predictions about optimal behavior in these foraging environments (with some caveats below about relevant features of the task design and analysis), I'm less certain about what novel insights it offers about the underlying neural mechanisms. Overall, I think more needs to be done to characterize behavior in this task, to describe the nature of the underlying value signals, and to determine what unique conclusions can be drawn about these circuits based on the current task.

The authors identify a network of regions that show multivariate encoding of socially-adjusted value, which they show is a key determinant of their foraging behavior. It's hard to know what to make of this finding, either in the context of these foraging decisions (in terms of what roles these regions play in this valuation process and why) or more generally (in terms of what these activations tell us about foraging and socially adjusted value in particular). First, in spite of its novelty, this multivariate approach obscures how exactly socially-adjusted value is being encoded by these regions. For instance, is it consistently being coded positively for the current patch and negatively for the alternate patch (in a way that might suggest a sort of reference-dependent relative value) or consistently coded positively in both (in a way that might suggest an overall value coding). (See also the comment below about how these values are being calculated, which is relevant to the current analyses that aim to distinguish RDMs for individual patch value and differences between patch values.) These patterns are relevant for thinking about what general processes are being indexed in these regions, since past work shows these same circuits involved with generally higher subjective value and increased likelihood/confidence of a particular choice. Some of these interpretations can generally be addressed/ruled out with response time data, but it appears in the current study that participants had to wait some time to respond so the critical response window may be lost on many trials. Univariate analyses (or parametric multivariate analyses like support vector regressions) would be useful for exploring these possibilities, and to substantiate the current implicit claim that the RDM approach is capturing qualitatively different patterns than univariate analyses would.

In terms of the behavioral findings, the overall patterns are compelling and consistent with the normative theory, but the models of the underlying decision process leave out some important elements. For instance, as far as I can tell, subjective value estimates currently don't incorporate learning of the patch values based on past experience (e.g., they assume that on the first trial participants have future value information) and they don't incorporate biases towards staying with the current patch. These additional elements could help to explain a larger share of the variance than they are currently able to, and potentially even account for nuances in the data such as the surprising finding in Figure S2 (where participants had the best collection rate at an intermediate level of competition in the safe condition, and monotonically lower collection rate with more competitors in the

threat condition). More importantly, these elements could also inform estimation and interpretation of neural activity related to decision value, confidence, uncertainty, etc.

Additional comments:

1. The overall readability of the paper could be substantially improved, to aid in understanding the task elements/dynamics and how value was calculated. For instance, Figure 1 is very difficult to parse and would benefit from additional/clearer labels to guide the reader through each panel (e.g., time-course of foraging, what the predation/competition optimal zones represent). It would also be helpful to expand (or supplement) Figure 1E with a plot of how choice varied as a function of safety and number of competitors. The description of the RDM approach and findings (pp. 9-10) is also somewhat opaque. Terminology should also be more consistent and clearly spelled out. For instance, survival value seems to be used interchangeably with socially adjusted value (if these are meant to refer to different things, that distinction should be clarified).

2. It wasn't entirely clear why participants engaged in repeating sequences (cycles) of patch options. As suggested above, this predictive component based on past learning could inform choice strategies and is worth considering when analyzing behavioral/neural data. For instance, do participants make consistent choices across cycles? Do neural correlates of value change over time as participants potentially rely more on their recollection of past choices?

3. The authors should include more detailed information about fMRI preprocessing parameters and what specific trial period regressors were modeled. It would also be worth clarifying how much overlap there is between the different trial elements that are being modeled (e.g., whether there is sufficient delay/jitter to distinguish the periods of interest), including how BOLD responses to the shocks themselves are being modeled separately from the remaining periods of interest (relevant particularly to the threat vs. safety map in Figure S3).

Reviewer #1 (Remarks to the Author):

The authors report an analysis of a new foraging paradigm in which participants are asked to choose a foraging patch while considering the density of competing agents and, under one condition, a predator that leads to a loss of tokens, a shock, and the end of the trial. Participants performed the task during functional magnetic resonance imaging (fMRI). After confirming that participants responded to the primary variables manipulated in the task (competitor density and predator threat), the authors construct a computational model of behavior modeling a concept called "socially adjusted value" which they show to be affected by points collected, probability of being shocked, and distaste for being shocked. The authors then test for drivers of variation in RDM matrices to identify areas of the brain that make the targeted distinction. They find a correlation for socially-adjusted value with putative value regions of the brain along with midcingulate cortex, posterior cingulate, and hippocampus and insula for the current patch. Representations for threat were found in similar regions. The authors conclude that a socially adjusted representation of value that incorporates threat and density tradeoffs is represented in a distributed network of brain regions.

The task is fun and thought provoking and the significant data collected for participants makes the paper more interesting as well as a set of complex analysis more feasible.

Thank you for these positive comments. In addition to our responses to the reviewer's specific comments, we wish to notify the reviewer that we have revised our modelling results to reflect the reviewer's comments. As a result, we have rerun the entire analysis with the revised estimates from a revised implementation of the model. While for the most part the results are the same as those reported in the original manuscript, this led to the OFC no longer being present in our RSA results for the survival value of the current patch. The results shown in Figure 2 reflect the updated analysis.

Larger concerns:

1. During the safe condition, it is clear from Fig S# that participants are still achieving high rates of reward with a larger number of competitors. It seems plausible that this is accomplished through faster response times and/or a greater number of key presses (though the details described are insufficient to be sure). Although there is a clear bias to choose the less populated patches, the choice is not absolute, this might be the reason. In general, the safe condition with more participants sounds just as rewarding and perhaps more engaging.

*Thank you for assisting us with ideas for this finding. After reviewing the comment and examining the plot we have adjusted the method used to extract competitor numbers across trials to create this figure. The aforementioned plot had used competitor numbers that existed at the **end** of all trials. While this is accurate for the short duration trials we also have longer trials (14s) during which the cycle changes (and hence the number of competitors change). On some occasions the participants acquired rewards after the cycle change. To address this we have extracted the exact timing for reward collection and recorded the number of competitors at that time, and not simply the end of the trial, which we had done to create the plot. A new plot has replaced the old in the manuscript. No patterns of consequence persist once this is done.*

More generally, we will also note that: 1) prior to playing the trial the participant does not know where the rewards will appear; 2) the participants were free to navigate in real time in each patch using the button box arrow keys; 3) the movement process is identical regardless of the number of competitors; 4) reward acquisition is also a function of the spawn location of the

reward vis a vis the location of the participant in the patch; and 5) the participant has only 1 goal after they are placed in the patch, namely to acquire the food rewards. These factors, in addition to the fact that the task does not encourage more movement in certain conditions, make it less likely that key presses would be different across trials.

If a larger number of movements (or more complex plans) are made it could explain the RSA result in the MCC as well as the widely distributed separability that hints at general motivation differences.

Our data suggest that for the vast majority of trials, little planning was utilized, evidenced by myopic choice dominance, that is, most decisions were made among the two visible patches, while few participants used the “later” choice options. The latter would require bringing forth representations of the next configuration in the patch cycle. In addition, there was little, if any, benefit to planning movements prior to starting trial, as the location of rewards was randomly determined on each trial. Instead, subjects had to behave reactively once the trial had begun. We are not able to say much about motivation once inside patch, since any given reward acquisition could be more due to spawn location vis a vis the player or, in other instances, a function of efficient movement, and in others, both aspects. But in any event, since the player has only one goal, reward acquisition, when inside the patch, we would be unlikely to find differences that motivation might make. However, motivational differences were observed in a different sense, with respect to reward vs. risk sensitivity, for which we can provide data – see figure S5, which is also included with caption, below.

Figure S5. Individual variability in competition difference between threat and conditions. Each bar represents an individual participant. Bar level indicates total competitors in the threat condition minus total competitors in the safe condition. High bars indicate extreme threat avoidance, while bars closer to 0 indicate no sensitivity to threat

2. Please add a description of the specification of ROIs. A quick reference to AAL appears but what was done was not clear from the methods. This also makes it difficult to compare these regions to the specific ROIs referenced in the manuscript.

Thank you for noting this. We have added the following full description of the ROIs to the methods section. We also wish to emphasize that these ROIs were only used for the purpose of illustrating the pattern of results in Figure 2, and all statistical inference was done at the whole-brain level. We have stated this more clearly within this new section in the methods.

“While statistical significance was determined using this whole-brain approach, for the purposes of illustrating the pattern of effects across different regions, we extracted beta values from the RSA within key regions showing significant effects (as shown in Figure 2). For this we used the AAL atlas, selecting the MCC, frontal medial orbital cortex, hippocampus and amygdala regions, extracting the mean beta value from the RSA within the region for each subject. We show the mean and 95% confidence intervals in Figure S5.”

We have also included an illustration of the ROIs embedded within Figure 2 to clearly illustrate their location, and we have included the ROIs as NIFTI images in the accompanying GitHub repository (see response to next point).

3. Degrees of freedom in t-tests change for similar tests without explanation.

Thanks for catching that typo. We adjusted the df, which is 23 for both tests reported in the behavioral results paragraph.

4. It is very challenging to review and have others replicate or extend work that is only available on request. Please: 1. Make all analysis scripts (main stats, SI, Neuroimaging, RSA) available on GitHub or a similar repository. 2. Make minimally processed fMRI data available on OpenNeuro or similar.

Thank you for this suggestion. We have placed all analysis scripts and behavioral data in a GitHub repository, and minimally processed fMRI data have been uploaded to OpenNeuro. We have added a section on data availability to the manuscript to direct readers towards these resources:

“All behavioral data and analysis code are available at https://github.com/mobbslab/foraging_paper. Neuroimaging data is available at <https://openneuro.org/datasets/ds003484>”

5. Need to see a correlation table (or list) between RDMs. Are they all independent?

The RDMs are largely independent, although there are some relatively high correlations, for example between difference scores and the individual variables used to calculate the difference score (maximum correlation = .65. We have included a correlation table in supplementary materials to show the exact statistics (Table S1, shown below). While this level of multicollinearity could be problematic with smaller numbers of observations, here our analyses involve thousands of observations as a result of being conducted on similarity matrices (i.e. each matrix has $n \text{ trials}^2$ observations), limiting its influence. However, to verify that parameter estimation is not problematic empirically, we have also run simulations viewable at https://github.com/mobbslab/foraging_paper/blob/master/notebooks/multicollinearity_test.ipynb to test parameter estimation accuracy with correlated predictors across different numbers of observations. These simulations indicated that even with highly correlated predictors and fewer observations than in our real analyses, parameters can still be estimated robustly (median correlation between true and estimated parameters = .996).

Table S1. Pearson correlations between representational dissimilarity matrices (RDMs) used in the RSA analysis. Values represent the mean correlation across subjects.

	Comp. diff	Current comp.	Alternative comp.	Threat	SV diff	Current SV	Alternative SV
Comp. diff							
Current comp.	0.65						
Alternative comp.	0.65	0.25					
Threat	-0.01	-0.02	0.00				
SV diff	0.21	0.18	0.20	0.21			
Current SV	0.09	0.19	0.00	0.67	0.49		
Alternative SV	0.11	-0.01	0.22	0.64	0.51	0.43	

Minor:

- line 27 "than" -> that

- line 51 "decoupled" usage unclear.

Changed to "independent"

- Social adjusted value always appears in separate font

- line 93 parenthetical contradicts immediately preceding statement without explanation.

Added / changed text to explain; the idea is that an individual's probability of capture is reduced in the presence of greater social density because of risk dilution.

- line 111 "The foregoing tendency to congregate in numbers in the presence..." is difficult to parse/understand

Edited text to clarify, same idea as comment re: line 93

- line 188 "based [on?] socially adjusted value"

- line 191-2 Difficult to understand. This is three things, right?

Yes, clarified

- line 205 "value" - disambiguate?

Clarified – socially adjusted value

- line 315 comma precedes citations at the end of the sentence

- line 325-6 font change.

The font changes appear to be an artifact of PDF conversion; fonts are consistent in the word doc. We will follow up with the Nature team.

- line 337 “contexts imbining the concept” - ? imbuing?

Yes!

- line 424-5 line break

- line 485 “will employ” - tense

Thank you for picking up on these issues. We have corrected them, and these corrections are highlighted in the revised value of the manuscript.

Reviewer #2 (Remarks to the Author):

The authors developed a naturalistic foraging task to measure how people evaluate competition during foraging with or without predation: When the environment is safe, an agent should want fewer competitors, but under threat, competitors offer safety from a predator. The authors used fMRI to look for multivariate representations of the value signals involved in making this choice. They found that participants adjusted their strategy depending on the threat context, and observed context-dependent value signals in ventromedial PFC and (posterior) mid-cingulate, suggesting that these regions are involved in integrating information about potential competition and threat to determine stay-leave decisions.

The topic of how people make decisions in dynamically changing social environments, and its connections to real-world foraging behavior, is of great interest to researchers across several disciplines. These authors have taken a novel approach to addressing this, using a cleverly designed task and substantial amounts of within-subject data. However, while I think this study has the potential to confirm predictions about optimal behavior in these foraging environments (with some caveats below about relevant features of the task design and analysis), I'm less certain about what novel insights it offers about the underlying neural mechanisms. Overall, I think more needs to be done to characterize behavior in this task, to describe the nature of the underlying value signals, and to determine what unique conclusions can be drawn about these circuits based on the current task.

We thank the reviewer for their positive comments.

The authors identify a network of regions that show multivariate encoding of socially-adjusted value, which they show is a key determinant of their foraging behavior. It's hard to know what to make of this finding, either in the context of these foraging decisions (in terms of what roles these regions play in this valuation process and why) or more generally (in terms of what these activations tell us about foraging and socially adjusted value in particular). First, in spite of its novelty, this multivariate approach obscures how exactly socially-adjusted value is being encoded by these regions. For instance, is it consistently being coded positively for the current patch and negatively for the alternate patch (in a way that might suggest a sort of reference-dependent relative value) or consistently coded positively in both (in a way that might suggest an overall value coding). (See also the comment below about how these values are being calculated, which is relevant to the current analyses that aim to distinguish RDMs for individual patch value and differences between patch values.) These patterns are relevant for thinking about what general processes are being indexed in these regions, since past work shows these same circuits involved with generally higher subjective value and increased

likelihood/confidence of a particular choice. Some of these interpretations can generally be addressed/ruled out with response time data, but it appears in the current study that participants had to wait some time to respond so the critical response window may be lost on many trials. Univariate analyses (or parametric multivariate analyses like support vector regressions) would be useful for exploring these possibilities, and to substantiate the current implicit claim that the RDM approach is capturing qualitatively different patterns than univariate analyses would.

Thank you for noting these points, which we agree are important with regard to the interpretation of our results. Our question here regards where these computed decision values are encoded, rather than how they are encoded. This is in line with the objective of using multivariate approaches such as RSA to characterize the information content of the signal, rather than its activity level (e.g. Kriegeskorte, Goebel & Bandettini, 2006, PNAS). Central to this approach is its ability to identify encoding of decision variables in the absence of any overall signal change, based on the distributed, multivariate pattern of activity within a given ROI. This is based on an implicit assumption that there the pattern of activity (perhaps indirectly representing varied activity levels within neuronal subpopulations in the ROI) should encode decision variables. As a result, there is not necessarily a clear directionality to the way these variables are encoded, as may be identified using univariate analyses. In return however, we gain the ability to identify regions that encode these variables in ways that are more complex and subtle than through overall activity levels. The reviewer also makes an important point regarding the potential involvement of higher level processes, such as confidence and subjective value, and it is indeed difficult to truly determine their influence on the results. The one aspect of our results that may address this to some extent relates to our coding of the RDMs for socially adjusted value – these represent high negative value as maximally dissimilar to high positive value. Thus, the results we show indicate that this variable is encoded on a continuous scale, rather than being coded as the absolute value. We have added the following to address this issue in detail in the discussion:

“While this approach does enable us to identify where in the brain these key decision variables are encoded, it provides little information about how these variables are encoded. Multivariate approaches rely on identifying distributed patterns of activity that are linked to a particular variable of interest, but it is challenging to qualitatively assess exactly what this encoding pattern looks like. For example, we are unable to say for certain that socially adjusted value is encoded relative to a particular reference value. Instead, this approach makes the implicit assumption that the variable of interest is encoded by the relative values of neural subpopulations within the region being examined. Thus, while simple coding schemes may exist within these small-scale populations, at the macro scale we cannot identify such a straightforward coding. Relatedly, there are other processes that may influence our results. For example, decision confidence and subjective value (which may be distinct from the value underpinning decisions) are likely to be confounded with socially adjusted value to some degree, but were not measured in this study. Further work will be required to understand how these processes interact to influence decision making and the subjective experiences of subjects during these tasks.”

We agree that reaction times would have been helpful in addressing these questions, however subjects did indeed have to wait before making a decision (to ensure a period of deliberation on which to focus the imaging analyses), preventing us from conducting these analyses. We have added this as a limitation in the discussion:

“While our results provide insights into the neural representation of the task presented here focused on neural representations at the cost of detailed behavioral measures. For example, because subjects were given a period in which to consider their decision before

having very little time to actually enter the decision (to enable analysis of the neural representation of the decision-making process), we do not have access to reaction time data, which may have allowed us to investigate factors such as decision confidence.”

We thank the reviewer for their suggestion regarding univariate analyses as a way to illustrate how the multivariate approach is capable of capturing qualitatively different effects to traditional univariate analyses. We have conducted these analyses and have summarized them in the results section, with full results added to supplementary material. As expected, these analyses identify largely distinct regions, indicating that our RSA analyses are indeed capturing qualitatively different patterns.

“Univariate analyses were conducted using Lyman (<http://www.cns.nyu.edu/~mwaskom/software/lyman/index.html>). Data were first smoothed using an 8mm FWHM kernel, and first level models were constructed as described for the representational similarity analysis for each run. The only exception was that rather than modelling the decision period of each trial separately, we instead modelled the decision period using a single regressor for each trial, including parametric modulators for each decision variable of interest (socially adjusted value and the number of competitors for the current and alternative patch, the difference between these values, and threat level). Maps from first level analyses for each run were then combined using a second-level fixed-effects analysis within subject, and subject-level maps from this stage were finally used for group-level analysis. At the group level, significance was determined using FSL’s randomize tool, with 5000 permutations and 10mm variance smoothing.” - Methods

“To complement the RSA results, we also performed univariate analyses to identify regions in which overall activity levels varied according to the key decision variables involved in the task (Figure S3). These analyses showed that a greater number of competitors in the current patch was associated with increased activity in visual cortex, while more competitors in the alternative patch were associated with reduced activity in visual cortex. This pattern in visual cortex also emerged when looking at the effects of socially adjusted value, however in addition greater value in the current patch was associated with greater activity in the thalamus, while greater value in the alternative patch was linked to greater activity in the OFC and mPFC and reduced activity in the thalamus, insula, and dorsal striatum. No significant clusters emerged when focusing on the effect of threat during the decision phase.” – Results

“Our use of RSA allowed us to focus on the multivariate representations of these variables in the brain, rather than relying on single-voxel activity changes... This is demonstrated by comparison to our univariate analyses, which do not identify the same patterns.”

In terms of the behavioral findings, the overall patterns are compelling and consistent with the normative theory, but the models of the underlying decision process leave out some important elements. For instance, as far as I can tell, subjective value estimates currently don’t incorporate learning of the patch values based on past experience (e.g., they assume that on the first trial participants have future value information) and they don’t incorporate biases towards staying with the current patch. These additional elements could help to explain a larger share of the variance than they are currently able to, and potentially even account for nuances in the data such as the surprising finding in Figure S2 (where participants had the best collection rate at an intermediate level of competition in the safe condition, and monotonically lower collection rate with more competitors in the threat condition). More importantly, these elements

could also inform estimation and interpretation of neural activity related to decision value, confidence, uncertainty, etc.

Thank you for this suggestion. We agree that it would be desirable to gain additional insights into the behavior during the task, however we are unfortunately limited by the design of the task itself, which was created to elucidate the neural mechanisms supporting the foraging decision itself and sadly provides limited opportunity for characterizing other components of the foraging process.

The participant is required to learn the patch dynamics, e.g. cycles, during training prior to beginning the actual task. We require the participant to verbally describe the patch evolution prior to starting the fMRI task. However, we found that subjects very rarely chose to use this information in their decision making. It may be that affective value not experienced (because training is not “live”) has some kind of influence.

It is possible however that subjects do learn the value of each patch throughout the task (i.e. learning the reward expected for each number of competitors) and use this to determine which patch to choose. Based on the reviewer’s suggestion, we tested a model that incorporated learning the value of each patch based on experience using a reinforcement learning model, allowing subjective expectations of reward from each patch to be formed as the task progresses. However, this performed marginally worse than our initial model that did not incorporate learning.

While this may seem surprising, we note that A) subjects were trained on the task prior to entering the scanner, providing an opportunity to learn how much reward they might expect from each patch prior to beginning the task in the scanner, and B) given the length of the task (4 hours in total), even if there were some learning at the beginning of the task it is likely that the vast majority of the task involves little learning. As a result, while this task is optimized for investigating the decision process, it is limited with regard to investigating learning, and the fluctuations in uncertainty that come with learning.

We also tried fitting a variant of the decision model that incorporated choice stickiness (i.e. a tendency to stick with the previously chosen option) by adding value to the previously chosen patch. However, this also did not provide as good a fit to the data as the original model. We have added a description of our model-fitting procedure and its results to the methods and results and have added a section on this issue to the discussion.

We also wish to notify the reviewer that in revising our modelling results we noted a minor error in the implementation of the original model. As a result, we have rerun the entire analysis with the revised estimates from a corrected implementation of the model. While for the most part the results are the same as those reported in the original manuscript, this led to the OFC no longer appearing in our RSA results for the survival value of the current patch. The results shown in Figure 2 reflect the updated analysis.

We have added the following text to describe this additional modelling:

“We also evaluated a model that learned the value of patches with different numbers of competitors and threat levels over the course of the task. On each trial, the expected value of the chosen patch was estimated using a Bayesian mean tracker model. The mean (m) and variance (v) of the patch value are updated on each trial using learning rate G as follows:

$$m_t = m_{\{t-1\}} + \delta_t \cdot G_t$$

$$v_t = (1 - G_t) \cdot v_{\{t-1\}}$$

The learning rate G is updated on each trial as a function of the variance and an additional free parameter theta (fixed at 1), representing the error variance.

$$G_t = \frac{v_{\{t-1\}}}{v_{\{t-1\}} + \theta_\epsilon^2}$$

Finally, we tested a variant of the initial decision model with an additional free parameter (S) representing a tendency to stick with the previously chosen patch rather than switching to the alternative. Here, the value of the chosen option from the previous trial was multiplied by S , which has the effect of making it more likely to be chosen.”

“We compared this model to variants including one that learned the value of different patches, depending on the number of competitors and threat level, and one that incorporated a tendency to stick with the previously chosen patch. Model comparison indicated that the initial model provided a better fit (WAIC = -6181.56, higher=better) versus the variant incorporating learning (WAIC = -6219.19) and the variant incorporating choice stickiness (WAIC = -9829.30).”

“As a result of our focus on the decision-making process, the task was also not ideally suited to investigate how subject may learn the value of different patches when faced with competition and threat, and how they may make decisions that incorporate uncertainty in these learned values. This is illustrated by our modelling analyses, in which we found that a model incorporating learning did not provide as good a fit to the data, accounting for model complexity. Rewards were stable over the course of the task, and subjects were able to practice the task prior to entering the scanner, allowing them to learn the value of different options prior to beginning the real task. Additionally, the length of the task (approximately 4 hours) means that even if some learning does occur early in the task, the majority of the task will involve little learning.”

Additional comments:

1. The overall readability of the paper could be substantially improved, to aid in understanding the task elements/dynamics and how value was calculated. For instance, Figure 1 is very difficult to parse and would benefit from additional/clearer labels to guide the reader through each panel (e.g., time-course of foraging, what the predation/competition optimal zones represent). It would also be helpful to expand (or supplement) Figure 1E with a plot of how choice varied as a function of safety and number of competitors.

We have updated the description of the RDM methods and results in the manuscript and rewritten parts of it to clarify the procedure (these are too numerous to include here but are highlighted in the revised manuscript on pages 10-11). Our use of “survival value” was inadvertently left in from a prior manuscript version, and we have updated all instances to ensure “socially-adjusted value” is used throughout.

We have reworked figure 1 to more clearly describe the task elements and dynamics, including the time course of trials. We have also adjusted the theoretical model to better capture the relationship between threat, competition and socially adjusted value. We have added the below plot to figure 1 to address the above comment re: 1E (which in the new figure will be 1D).

A: Task Design. In the safe play (blue patches) phase the player 1) observes possible patches; 2) views the decision options; 3) makes a patch decision; and 4) is placed into the selected patch then competes to capture rewards; in the danger phase (red patches) the player follows the same procedure as safe play, except that the participant is subject to potential capture by the predator. In the safe condition the trial ends after time expires. The side-by-side patches were the same color during the experiment and are just color-coded here to clarify the differences between safe and threat conditions.

D: Decision tendency as a function of threat and competition. Participants selected patches with few competitors more frequently in the safe condition, and patches with more competitors in the threat condition.

The description of the RDM approach and findings (pp. 9-10) is also somewhat opaque. Terminology should also be more consistent and clearly spelled out. For instance, survival value seems to be used interchangeably with socially adjusted value (if these are meant to refer to different things, that distinction should be clarified).

2. It wasn't entirely clear why participants engaged in repeating sequences (cycles) of patch options. As suggested above, this predictive component based on past learning could inform choice strategies and is worth considering when analyzing behavioral/neural data. For instance, do participants make consistent choices across cycles? Do neural correlates of value change over time as participants potentially rely more on their recollection of past choices?

The design had contemplated decisions that could take advantage of current viewable stimuli (presented patches) or, for a subset of trials, another set of patches (the next set in the patch sequence). Thus the participant could select a patch that they could literally see, or the left or right patch of the next set in the cycle. However, it turned out that participants almost never selected "later" patches (selecting this option on approximately 7.5% of trials), even though they were trained on the sequences and were required to pass a verbal quiz in order to proceed to the live trials. Thus, participants had evidently learned the sequences and played practice trials prior to engaging in the live task, thereby further strengthening the knowledge acquired during initial training. Since we didn't foresee such little use of the "later" option, we do not have a clear explanation for why it was used so infrequently. It is possible there was too much cognitive load to consider the current and next patch set options, or perhaps the perceived difference wasn't significant enough to put forth the cognitive effort. As a result we opted to disregard this component of the task in our analyses as there would not be a sufficient number of trials to compare these decisions versus immediate decisions.

3. The authors should include more detailed information about fMRI preprocessing parameters and what specific trial period regressors were modeled. It would also be worth clarifying how much overlap there is between the different trial elements that are being modeled (e.g., whether there is sufficient delay/jitter to distinguish the periods of interest), including how BOLD responses to the shocks themselves are being modeled separately from the remaining periods of interest (relevant particularly to the threat vs. safety map in Figure S3).

Thank you for noting this. Regarding the preprocessing parameters, we have now included the standard description of the fMRIPrep pipeline provided by the creators of the tool, which details the steps used.

We also appreciate that the issue of signal contamination by shock administration is an important one. The task was designed such that shock administration did not occur close to the decision period (being separated by at least 6 seconds). In addition, shocks were relatively rare (being administered on 4.6% of trials). We have added the following to address this concern:

"The trial finished with a 3 second ITI before the next trial began. Importantly, the combination of the 3 second ITI and 3 second period where options were displayed (prior to available options on the current trial being highlighted) ensured that the decision period (the focus of the imaging analyses) was separated from the administration of shock by at least 6 seconds."

We have also revised the description of the first-level models to describe which trial elements were modelled:

“As our analyses focused on the decision period of the task, we modelled this period of each trial using a separate regressor, resulting in independent beta maps for each trial³⁴. Aside from the decision period, we also modelled the pre-decision period, between the trial starting and the available options being revealed, the foraging period, and the receipt of shock as events with durations corresponding to their duration in the trial, convolved with the standard hemodynamic response function. These were modelled using single regressors representing all trials across the run.”

REVIEWER COMMENTS

Reviewer #1 (Remarks to the Author):

I definitely would have loved to see analysis of reaction time data. It would be very useful for interpreting the mental state of the participants (as echoed in R2s comments). With that simply not possible, I am happy to see the manuscript be further interpreted by the broader community. The authors have sufficiently addressed my concerns.

Reviewer #2 (Remarks to the Author):

The authors have addressed several of my previous comments and the manuscript has improved as a result. Overall, I continue to think that this is a novel and compelling approach and that the behavioral results are interesting irrespective of the specific framing. However, I also continue to have the same general concern about the fMRI data, that in spite of the novel approach it's not clear whether neural correlates identified by this task provide unique insight into values that are socially-adjusted rather than reflecting value in a general sense (i.e., what would be seen in more basic choice tasks). The new univariate analyses and additional caveats in the Discussion section are helpful on this front, but end up concluding more or less that it can't be known what the multivariate signals are reflecting. This felt unsatisfying and I think can be helped with a few additional analyses that can provide a bit more insight and finality to the results.

First, in terms of the estimation of socially-adjusted value (SAV), the model currently seems to fall short of addressing the interesting question at the heart of this paper: how do decision-makers use information about competition to adjust value? A generative model of such a social decision should ideally integrate information about the number of competitors and threat level to produce choice (and particularly the non-linear relationship predicted by theory). I think this type of model revision would ultimately be more appealing and informative than the current one, and would license stronger claims about a behavioral and neural measure of the social adjustment process. However, if they prefer to continue to use the current model (which carries more of a model-free or episodic estimate of points that would be accrued in a given condition, seemingly agnostic to social information per se) then I would simply recommend re-naming the variable to avoid the 'socially-adjusted' modifier and re-framing the relevant interpretations more conservatively to refer to the contribution of 'subjective value' signals in the brain and choice, rather than making claims about the social adjustment of value.

Second, in terms of the univariate analyses, these GLMs (like all of the earlier ones in the paper) anchor on the current vs. alternative patch and the difference between them, as though this is the main frame of reference for the participant. It's possible that this wasn't the case, especially since there wasn't a clear bias towards the previous patch in behavior. I would suggest performing additional analyses that instead focus on common neuroeconomic variables for choices in general:

- (a) SAV for the chosen patch
- (b) SAV for the unchosen patch
- (c) Chosen - unchosen SAV
- (d) |Chosen - unchosen SAV|
- (e) Chosen + unchosen SAV

I would recommend at least testing c-e (which may require separate GLMs depending on collinearity) to provide a clearer comparison with commonly observed value signals and provide additional opportunities for aligning univariate and multivariate patterns of activation (though these of course may not align ultimately, for reasons the authors note).

Whatever the result of these analyses, I think the positive points raised above and in my previous review still make this paper of interest to a broad audience. But I think these kinds of analyses will help to substantiate or mitigate claims related to the specificity of the overall neural findings to the current task/domain.

Reviewer #1

I definitely would have loved to see analysis of reaction time data. It would be very useful for interpreting the mental state of the participants (as echoed in R2s comments). With that simply not possible, I am happy to see the manuscript be further interpreted by the broader community. The authors have sufficiently addressed my concerns.

Thank you, we are glad to have been able to address the concerns raised. We agree regarding the insights that could be generated by reaction time data, and this will be something we address in future work.

Reviewer #2

The authors have addressed several of my previous comments and the manuscript has improved as a result. Overall, I continue to think that this is a novel and compelling approach and that the behavioral results are interesting irrespective of the specific framing. However, I also continue to have the same general concern about the fMRI data, that in spite of the novel approach it's not clear whether neural correlates identified by this task provide unique insight into values that are socially-adjusted rather than reflecting value in a general sense (i.e., what would be seen in more basic choice tasks). The new univariate analyses and additional caveats in the Discussion section are helpful on this front, but end up concluding more or less that it can't be known what the multivariate signals are reflecting. This felt unsatisfying and I think can be helped with a few additional analyses that can provide a bit more insight and finality to the results.

Thank you for these suggestions.

1) First, in terms of the estimation of socially-adjusted value (SAV), the model currently seems to fall short of addressing the interesting question at the heart of this paper: how do decision-makers use information about competition to adjust value? A generative model of such a social decision should ideally integrate information about the number of competitors and threat level to produce choice (and particularly the non-linear relationship predicted by theory). I think this type of model revision would ultimately be more appealing and informative than the current one, and would license stronger claims about a behavioral and neural measure of the social adjustment process. However, if they prefer to continue to use the current model (which carries more of a model-free or episodic estimate of points that would be accrued in a given condition, seemingly agnostic to social information per se) then I would simply recommend re-naming the variable to avoid the 'socially-adjusted' modifier and re-framing the relevant interpretations more conservatively to refer to the contribution of 'subjective value' signals in the brain and choice, rather than making claims about the social adjustment of value.

A: This is a very good point, and we agree that a more sophisticated generative model would allow us to make stronger inferences regarding the computation of socially adjusted value. Having considered how such a model might be implemented, we concluded that the task design is such that it would not be possible to determine with any certainty whether such a strategy was indeed being used by subjects. To assess this properly, the task would need to include conditions where the socially adjusted value can be calculated (based on the expected reward and number of competitors) but not learned in a model-free manner. In our task, the value of each condition could be learned prior to beginning the task, making it difficult to determine for certain that an integrative value calculation was taking place, no matter how sophisticated the model. Given this limitation, we agree with the reviewer that the variable naming should be changed.

We have changed "socially-adjusted value" to "socially-dependent value", which we believe retains the importance of the social component of the task without implying that an existing non-social value estimate is adjusted based on the level of competition. We did attempt to rewrite the manuscript referring to "subjective value" instead but found that this obscured the critical contribution of the paper,

namely how subjective value depends upon social context. We have modified the introduction to introduce this term more clearly:

“Accordingly, optimal foraging theories^{1,2} suggest that the conflicting trade-off between threat of competition and the threat of predation are mitigated by the overall fitness or socially-dependent value of the patch. That is, subjective value should depend upon not only the level of reward or threat in the environment, but also the social context.”

And have removed references to the calculation of socially adjusted value:

“one hypothesis is that these regions reflect the ~~socially-adjusted~~ **socially-dependent value** of foraging decisions ~~through the integration of knowledge about competition and threat, constructing,~~ **providing** a primary decision variable more immediately relevant to behavior than the simple social density of an environment”

“suggesting that these regions ~~integrate information about social density and threat to calculate~~ **represent** the overall ~~socially-adjusted~~ **socially-dependent** value of both the current patch and an alternative”

We have also included the following paragraph in the discussion to temper our conclusions regarding the social “adjustment” of subjective value:

“An important caveat of this work is that we cannot be certain how subjects determine the socially-dependent value of a patch. In our task, it was possible for subjects to learn the value of each patch in a model-free manner, and this was reflected in our computational modelling. As a result, while the value of a patch clearly depends on the social component of the task, the subject does not necessarily need to perform any online integrative computations that adjust the value of a patch based on the level of competition, even if this may be a more effective strategy in the real world. To investigate this further, future work will need to exploit more complex experimental designs in which socially-dependent value cannot be learned easily.”

2) Second, in terms of the univariate analyses, these GLMs (like all of the earlier ones in the paper) anchor on the current vs. alternative patch and the difference between them, as though this is the main frame of reference for the participant. It’s possible that this wasn’t the case, especially since there wasn’t a clear bias towards the previous patch in behavior. I would suggest performing additional analyses that instead focus on common neuroeconomic variables for choices in general:

(a) SAV for the chosen patch

(b) SAV for the unchosen patch

(c) Chosen - unchosen SAV

(d) |Chosen - unchosen SAV|

(e) Chosen + unchosen SAV

I would recommend at least testing c-e (which may require separate GLMs depending on collinearity) to provide a clearer comparison with commonly observed value signals and provide additional opportunities for aligning univariate and multivariate patterns of activation (though these of course may not align ultimately, for reasons the authors note).

Whatever the result of these analyses, I think the positive points raised above and in my previous review still make this paper of interest to a broad audience. But I think these kinds of analyses will help to substantiate or mitigate claims related to the specificity of the overall neural findings to the current task/domain.

A: Thank you for this suggestion, it is indeed possible that subjects' main frame of reference was not the focused on the current versus alternative decision. We have run the univariate analysis suggested (using separate GLMs due to collinearity) and have summarized the results in the main manuscript, with full results shown in the supplementary material. Briefly, these analyses primarily showed generally negative effects of subjective value for the chosen option (both in isolation and relative to the unchosen option) in dorsolateral prefrontal cortex, anterior insula, and dorsal anterior cingulate cortex. This suggests these regions were more active when the chosen option had lower value than the unchosen option (i.e. when was likely to result in getting caught). There was no evidence for activity associated with the absolute difference, or the total value of the options. These results contrast with those typically observed in traditional value-based decision-making tasks, which often find chosen-unchosen value difference is coupled to activity in ventromedial prefrontal cortex. However we believe this can be explained by the fact that A) this is not a traditional task in the sense that it does entail a foraging component (even if this is presented to the subject less explicitly than is often the case in foraging tasks, and B) the outcomes involve threat, making negative value far more salient than in typical tasks using monetary gain/loss.

We have summarized these results in the revised manuscript and added a figure showing the results to the supplementary material.

“Additional analyses focusing on the value difference between chosen and unchosen options were conducted using the same approach, including regressors representing the value of the chosen option, the value of the unchosen option, the value difference between chosen and unchosen options, the absolute value difference between chosen and unchosen options, and the sum of the value of the two options. These were modelled in separate GLMs due to collinearity between regressors.”

“Finally, to facilitate comparison with prior work on value-based decision-making, we performed univariate analyses focusing on the difference in value between the chosen and unchosen patch (in contrast with our primary analyses, which focused on the current and alternative patch). Results are shown in Figure S6, and these revealed widespread negative effects of chosen – unchosen value difference, including clusters in the dorsal anterior cingulate cortex and dorsolateral prefrontal cortex, with a single cluster in the right dorsolateral prefrontal cortex associated with the absolute value difference. Activity associated with the value of the chosen and unchosen option independently was observed primarily in occipital areas.”

Figure S6. Results of univariate analyses focused on value of the chosen and unchosen patches. Values represent t-statistics and maps are thresholded at $p < .05$ TFCE corrected. SDV: Socially dependent value.

REVIEWER COMMENTS

Reviewer #2 (Remarks to the Author):

I appreciate the extensive additional univariate analyses that the authors performed, and I think readers will find these results valuable.

However, I was disappointed by their response to the first main concern I raised. Replacing 'socially-adjusted value' with 'socially-dependent value' doesn't address the core issue, which is that the authors don't provide any evidence that these value estimates rely on a consideration of social variables. Their model learns what rewards to expect in a given patch, but it does so blind to any social information. If the experiment had included patches without competition, the model would have learned value estimates in the same way, and the value estimates it produces (e.g., as inputs to the fMRI analyses) would not differentiate between socially-dependent patch values and socially-independent patch values (i.e., foraging in isolation). Similarly, none of the model's features lend themselves to generalizing to other socially-dependent contexts any more than to contexts involving foraging in isolation.

It's still reasonable for the authors to contextualize the paper's relevance and predictions in terms of the dependence of foraging decision on social context (which is a large part of what makes this research novel and of broad interest). However, I think using the phrase 'socially-dependent' when referring to what was explicitly estimated by the model and measured in the brain has the potential to mislead. My recommendation would be to instead refer to this experimental variable as 'patch values' or 'subjective patch values,' which is specific enough to be clear what the authors are referring to and how it relates to the experimental paradigm, but avoids implying that these values are computed in a socially-dependent fashion. The future modeling and experimental work suggested by the authors could nevertheless go on to justify this additional specificity.

To expedite remaining revisions, here are examples of more specific recommendations along these lines (though I defer to the editor and authors in terms of specific language):

"We formulate a mathematically grounded quantification of socially-dependent value in social foraging environments and show using multivariate fMRI analyses that socially-dependent value is encoded by mid-cingulate and ventromedial prefrontal cortices, regions that integrate action and value signals."
 "We formulate a mathematically grounded quantification of patch value in social foraging environments and show using multivariate fMRI analyses that such value is encoded by mid-cingulate and ventromedial prefrontal cortices, regions that integrate action and value signals."

"In a task containing changing levels of competition in safe and hazardous patches with virtual predators, we demonstrate that human participants inversely select competition avoidant and risk diluting strategies depending on socially-dependent [sic]. We formulate a mathematically grounded quantification of socially-dependent value in social foraging environments and show using multivariate fMRI analyses that socially-dependent value is encoded by mid-cingulate and ventromedial prefrontal cortices, regions that integrate action and value signals. Together, these results suggest that these cortical regions play a role in adjusting to both competitive and predatory threats."

→ "In a task containing changing levels of competition in safe and hazardous patches with virtual predators, we demonstrate that human participants inversely select competition avoidant and risk diluting strategies depending on the learned patch value. We formulate a mathematically grounded quantification of this patch value in social foraging environments and show using multivariate fMRI analyses that patch value is encoded by mid-cingulate and ventromedial prefrontal cortices, regions that integrate action and value signals."

For related reasons, the authors should avoid making strong mechanistic claims about how their results generalize to neural representations involved in broader social contexts (e.g., where patch conditions are not extensively trained and individuals have to integrate information about competitors

and other social variables on the fly). For example, I would recommending toning down or removing this section from the discussion:

“These results indicate that when making decisions about foraging patches in the presence of competitors, the brain predominantly represents the socially-dependent value of a potential patches, regardless of whether this represents risk of predation or risk of competition for food, while potentially disregarding the observed number of competitors, or the difference between available patches. Importantly, this suggests that in social foraging environments the brain is representing the value of social density in terms of a socially-dependent value, rather than simply representing the numbers of competitors, and suggests that the brain encodes the value of independent options rather than their difference. Many of the regions encoding socially-dependent value, such as MCC and vmPFC, also encoded overall threat level (i.e. at risk of predation or safe). This may indicate that these same regions simultaneously encode threat present in the environment generally and the socially-dependent value of potential patches.”

REVIEWER COMMENTS

Reviewer #2 (Remarks to the Author):

I appreciate the extensive additional univariate analyses that the authors performed, and I think readers will find these results valuable.

We thank the reviewer for taking the time to clearly explain the concerns highlighted below and the specificity and guidance to improve the quality of the manuscript.

We agree that these analyses will provide a clearer and fuller picture of the phenomenon we describe in the manuscript.

However, I was disappointed by their response to the first main concern I raised. Replacing ‘socially-adjusted value’ with ‘socially-dependent value’ doesn’t address the core issue, which is that the authors don’t provide any evidence that these value estimates rely on a consideration of social variables. Their model learns what rewards to expect in a given patch, but it does so blind to any social information. If the experiment had included patches without competition, the model would have learned value estimates in the same way, and the value estimates it produces (e.g., as inputs to the fMRI analyses) would not differentiate between socially-dependent patch values and socially-independent patch values (i.e., foraging in isolation). Similarly, none of the model’s features lend themselves to generalizing to other socially-dependent contexts any more than to contexts involving foraging in isolation.

It’s still reasonable for the authors to contextualize the paper’s relevance and predictions in terms of the dependence of foraging decision on social context (which is a large part of what makes this research novel and of broad interest). However, I think using the phrase ‘socially-dependent’ when referring to what was explicitly estimated by the model and measured in the brain has the potential to mislead. My recommendation would be to instead refer to this experimental variable as ‘patch values’ or ‘subjective patch values,’ which is specific enough to be clear what the authors are referring to and how it relates to the experimental paradigm, but avoids implying that these values are computed in a socially-dependent fashion. The future modeling and experimental work suggested by the authors could nevertheless go on to justify this additional specificity.

Given these detailed, specific, and helpful comments, we understand the reviewer’s concern regarding using the term “social” to describe the phenomenon associated with our modeling. After careful consideration of various alternatives, we have chosen the term “perceived patch value” (PPV). This term allows for interpretation that value is subjective and therefor may be comprised of multiple components that are assessed and weighed differently between individuals.

To expedite remaining revisions, here are examples of more specific recommendations along these lines (though I defer to the editor and authors in terms of specific language):

“We formulate a mathematically grounded quantification of socially-dependent value in social foraging environments and show using multivariate fMRI analyses that socially-dependent value is encoded by mid-cingulate and ventromedial prefrontal cortices, regions that integrate action and value signals.”

 “We formulate a mathematically grounded quantification of patch value in social foraging environments and show using multivariate fMRI analyses that such value is encoded by mid-cingulate and ventromedial prefrontal cortices, regions that integrate action and value signals.”

“In a task containing changing levels of competition in safe and hazardous patches with virtual predators, we demonstrate that human participants inversely select competition avoidant and risk diluting strategies depending on socially-dependent [sic]. We formulate a mathematically grounded quantification of socially-dependent value in social foraging environments and show using multivariate fMRI analyses that socially-dependent value is encoded by mid-cingulate and ventromedial prefrontal cortices, regions that integrate action and value signals. Together, these results suggest that these cortical regions play a role in adjusting to both competitive and predatory threats.”

→ “In a task containing changing levels of competition in safe and hazardous patches with virtual predators, we demonstrate that human participants inversely select competition avoidant and risk diluting strategies depending on the learned patch value. We formulate a mathematically grounded quantification of this patch value in social foraging environments and show using multivariate fMRI analyses that patch value is encoded by mid-cingulate and ventromedial prefrontal cortices, regions that integrate action and value signals.”

Thank you for the specificity – we have adjusted the manuscript based on our new term accordingly and throughout the manuscript.

For related reasons, the authors should avoid making strong mechanistic claims about how their results generalize to neural representations involved in broader social contexts (e.g., where patch conditions are not extensively trained and individuals have to integrate information about competitors and other social variables on the fly). For example, I would recommend toning down or removing this section from the discussion:

“These results indicate that when making decisions about foraging patches in the presence of competitors, the brain predominantly represents the socially-dependent value of a potential patches, regardless of whether this represents risk of predation or risk of competition for food, while potentially disregarding the observed number of competitors, or the difference between available patches. Importantly, this suggests that in social foraging environments the brain is representing the value of social density in terms of a socially-dependent value, rather than simply representing the numbers of competitors, and suggests that the brain encodes the value of independent options rather than their difference. Many of the regions encoding socially-dependent value, such as MCC and vmPFC, also encoded overall threat level (i.e. at risk of predation or safe). This may indicate that these same regions simultaneously encode threat present

in the environment generally and the socially-dependent value of potential patches.”

We adjusted the section to concur with the overall tenor of the comments:

These results indicate that when making decisions about naturalistic foraging patches (i.e. in which several variables factor into decision-making), the brain represents the perceived patch value of potential patches, a computational result that incorporates multiple streams of information, but does not appear to explicitly include the value difference between available patches. this While our modeling approach is agnostic to the social component per se, social density may be represented in terms of a perceived patch value, rather than simply the numbers of competitors. Accordingly, the brain may encode the value of independent options rather than their difference. Many of the regions encoding perceived patch value, such as MCC and vmPFC, also encoded overall threat level (i.e. at risk of predation or safe). This may indicate that these same regions simultaneously encode threat present in the environment generally and the perceived patch value of potential patches.